# Acute Effect of Kettlebell Swings on Sprint Performance

**DOI:** 10.3390/sports7020036

**Published:** 2019-02-10

**Authors:** Kishen Kartages, Guy C. Wilson, Che Fornusek, Mark Halaki, Daniel A. Hackett

**Affiliations:** Discipline of Exercise and Sport Science, The University of Sydney, Sydney, NSW 2141, Australia; kkar4118@uni.sydney.edu.au (K.K.); guy.wilson@sydney.edu.au (G.C.W.); che.fornusek@sydney.edu.au (C.F.); mark.halaki@sydney.edu.au (M.H.)

**Keywords:** preconditioning exercise, post-activation potentiation, power, sprint running

## Abstract

Previous research has shown that kettlebell swings (KBS), utilizing the hip-hinge technique, exhibit similar lower-limb muscle activation patterns to sprint running. This study investigated whether the inclusion of KBS in the warm-up enhances sprint performance. Moderately trained males (*n* = 12) and females (*n* = 8) performed KBS and a control (CON) condition (passive rest) in random order before performing three 20-m sprint trials separated by 4 min. No condition (KBS versus CON) effects, time effects or condition by time interactions were found for sprint times at 5-m and 10-m. A significant time effect was found for sprint time at 20-m with faster sprint time at 12 min compared to 4 min (*p* = 0.022). No condition effect or condition by time interaction was found for sprint time at 20-m. Small to moderate correlations were found for change in sprint time (CON minus KBS) and KBS load at 4, 8, and 12 min. It appears the KBS is not effective for potentiating 20-m sprint performance; however, any potential benefit from the inclusion of KBS as a preconditioning exercise for sprinting may be influenced by individual strength capabilities with KBS.

## 1. Introduction

Preconditioning exercises, consisting of high resistance or ballistic activities, can be utilized by athletes in the minutes prior to power-based exercises in order to improve performance [1,2,3,4]. Studies have shown acute improvements in countermovement vertical jump height [3,5,6] and 20-m sprint [7,8,9,10] following resistance exercises such as barbell squats and power cleans. An acute enhancement in exercise performance following a preconditioning exercise is attributed to the post-activation potentiation (PAP) phenomenon, in which muscular performance characteristics are acutely enhanced as a result of their contractile history [11]. There have been two main theories proposed to explain an enhancement in athletic pursuits from PAP. These include (1) increased stimulation of the central nervous system, after preconditioning exercise, prior to an explosive activity [11,12] and (2) increased phosphorylation of myosin regulatory light chains within muscle fibres [12]. Both these theories explain an increase in contraction velocity of the potentiating muscle, thus leading to an increase in its power output during an explosive activity such as jumping, throwing, and sprinting.

Contrary to the findings of PAP effects on subsequent exercise performance, some studies have reported that acute exercise performance did not change [13,14] or was impaired [15] following a preconditioning exercise. An explanation for the conflicting results is likely due to the coexistence of potentiation and fatigue within the contracting muscle. For there to be an improvement in muscle performance after a preconditioning exercise, the potentiating effect must dissipate at a slower rate than the effect of fatigue. Findings from two meta-analyses [16,17] showed that individual strength profiles influence the rest period needed to induce PAP. Stronger compared to weaker individuals required shorter rest intervals (3–7 versus 6–12 min) with the former also having a longer potentiation duration (up to 12 min). An explanation for the shorter rest period required to induce PAP among stronger individuals is likely due to their greater fatigue resistance to heavier loads, greater distribution of type 2 muscle fibers and prior resistance training experience [13,18,19]. 

When using a resistance exercise to potentiate an athletic task, the effectiveness of the intensity and volume prescribed also appears to be influenced by muscular strength. The potentiating effect has been shown to be larger for stronger individuals when higher intensities are used (>85% of 1RM), and larger for weaker individuals when moderate intensities are used (60–84% of 1RM). In terms of training volume, stronger individuals express greater potentiation levels after a multiple set protocol, while weaker individuals benefit more from a single set protocol [16,17]. This suggests that for stronger individuals an increase in PAP from single to multiple sets outweighs the increase in fatigue. Therefore, individual characteristics (e.g., muscular strength and training experience) and various aspects of the exercise protocol used such as rest period, intensity and volume may influence the PAP effect from preconditioning exercises [12]. Previous studies that have investigated the effects of preconditioning exercises on sprint performance have used ballistic exercises such as depth jumps and alternate leg bounding [20,21] or resistance exercises such as barbell squats and power cleans [6,7,8,9,10,22,23,24,25]. The majority of these studies have shown improved sprint performance following the ballistic exercises [20,21] and resistance exercises [6,7,8,9,10,25]. However, it remains uncertain which type of exercise is more effective for maximizing the potentiating effect on sprinting [2,16]. Whereas the ballistic exercises require minimal equipment (e.g., weight vest) or no equipment, the performing of resistance exercises requires specialized equipment (e.g., barbells, weight plates, squat rack), which is generally inaccessible immediately prior to competitions [2,23]. To date, no study has investigated whether the kettlebell swing (KBS), in which repetitions are performed rapidly, as a preconditioning exercise can enhance sprint performance. Kettlebells are portable and can be easily transported; therefore, kettlebell exercises may be a suitable alternative to barbell exercises to induce a potentiating response to improve sprint performance. Furthermore, KBS with moderate loads have been shown to produce similar power outputs to jump squats using heavier loads [26] and to initiate higher proportion of horizontal to vertical propulsive forces compared to barbell squats and jump squats [27,28]. As such, KBS could potentially have a higher transferability to sprint running than squats. This would benefit track and field athletes more so than athletes involved in other sports (e.g., soccer, basketball, rugby) since the PAP effects will only last a short period of time. Whilst the PAP effect is generally observed with relative loads ≥ 60% 1RM [17], the loads used with KBS are typically prescribed based on a specific absolute load and have not exceeded 32 kg [26,29,30,31,32]. One possible reason for not expressing KBS loads based on a % 1RM could be due to the manner KBS are performed i.e., rapidly, which is not suited for 1RM testing. Therefore, to allow for a KBS load to be standardized according to an individual’s physical capabilities, the load could be determined based on highest mean power output. The primary aim of this study was to investigate whether including KBS (utilizing the hip hinge technique) in the warm-up enhances sprint performance in moderately trained males and females. A secondary aim was to examine whether a more pronounced PAP effect is observed in participants with greater leg strength and power. We hypothesized that sprint time would improve after performance of KBS compared to a control condition involving a passive rest. Also, we hypothesized that the PAP effect will be larger among participants with greater leg strength and power. 

## 2. Materials and Methods

### 2.1. Participants

Twenty healthy adults (12 males; 8 females), participated in the study. The sporting background of the participants included soccer, rugby, basketball, track and field, and martial arts. Characteristics of the participants are reported in Table 1. The inclusion criteria for this study included being aged 18–30 years, regularly performing at least 150 min of moderate exercise per week or 75 min of vigorous exercise per week, and having a minimum six months resistance training experience. Each participant was risk assessed via the use of the American College of Sports Medicine pre-exercise screening questionnaire and were deemed to be healthy. An information statement explaining all the procedures, benefits and risks of the study were given to the participants, as well as being verbally explained before commencement. Verbal and written informed consent was provided by the participants prior to commencing the study. The study protocol was approved by the University of Sydney Human Research Ethics Committee. 

### 2.2. Preliminary Sessions

Two preliminary sessions were conducted during the first two weeks (one session each week) (Figure 1). Each session involved anthropometric measures (height and weight), leg extensor strength and power testing, KBS practice and testing, and sprint practice. The best performance from the two preliminary sessions was used for the leg extensor strength and power, and KBS performance. On a separate visit during the second week body composition (i.e., lean body and fat mass) was assessed.

Leg strength and power of the knee extensors was assessed via isometric torque and isokinetic power respectively, using the Biodex isokinetic (Biodex System Pro 2, New York, USA) dynamometer. For assessment of isometric torque, participants completed 1 set of 3 maximum voluntary contractions (MVC) for each leg at a knee angle of 90 degrees flexion. Contractions were held for five seconds with 40 s rest between each contraction and 60 s rest between each leg. The assessment of isokinetic power involved participants completing 2 sets of 5 maximal concentric reciprocal contractions at two different angular velocities (90 °/s and 250 °/s). There was 60 s rest between sets and 120 s rest between each leg. Prior to testing, participants became familiar with both testing procedures by performing submaximal contractions for each muscle group as a warm-up. The best results for each leg were added together to calculate isometric torque and isokinetic power of the leg extensors. 

Participants were instructed and practiced the KBS with a hip dominant movement where emphasis was placed on maximal hip recruitment, and minimal knee flexion [26,33]. This technique was incorporated due to the generation of high horizontal propulsive forces, and increased activation of the posterior leg musculature corresponding with sprint running [26,27,34]. Mean power from KBS was measured via a linear force transducer (Gym Aware Power Tool, Canberra, Australia) attached to the kettlebell clamp. This was done via participants performing two swings where the mean power from the second swing was recorded, with loads progressing from light to heavy (increments of approximately 5–10 kg). There was approximately 30 s rest between trials for lighter loads and 60 s rest for heavier loads, based on participants’ perceived exertion. Once there was a decline in mean power output detected, no further increases in load were performed and the previous load (i.e., producing the highest mean power output) was used for the KBS in the experimental sessions. Sprint time was measured using wireless timing gates (Timing Solutions, Victoria, Australia) and were set up at 0-m, 5-m, 10-m and 20-m positions. Participants commenced the sprint following a three-second countdown and from a three-point stance with the participant’s foot and hand placed 50 cm behind the first timing gate. This ensured the participants commenced from a standardized position and did not trigger the gate prematurely. Data provided from the timing gate system was sprint time to each gate. Sprints were performed on a concrete surface under cover and shielded from winds (i.e., area between buildings). Participants were familiarized with sprinting using a three-point stance during the preliminary sessions. Five sprint attempts were given to each participant during each preliminary session and feedback on their three-point sprint start was provided by a co-investigator (K.K). Sprints commencing with a three-point stance has been used in previous studies investigating the PAP effect on sprint performance [7,24] and has been shown to be highly reliable (ICC = 0.92) [35].

A whole-body dual energy x-ray absorptiometry scanner (Lunar Prodigy, GE Medical Systems, Wisconsin, USA) was used to measure body composition. Scans were performed under standardized conditions (early morning, overnight fasted, and standardized body positioning on the scanning bed), by co-investigator (G.C.W). Inter-rater reliability based on scans at baseline was excellent for lean body and fat mass (ICC: 0.98–0.99 and CV: 1.1–2.4% respectively). Lean body and fat mass was determined using the system’s software package enCORE 2011 (version 13.60.033, GE Medical Systems, Wisconsin, USA).

### 2.3. Experimental Sessions

A repeated measures, crossover, experimental design with random treatment order, was used to investigate the acute effects of KBS on sprint performance (Figure 1). This study design was used to minimize the effects of bias (e.g., allocation and performance bias) and confounders (i.e., known and unknown). In week three, participants attended two experimental sessions separated by at least 48–72 h. One of the experimental sessions involved KBS and the other session a passive rest-control (CON) condition prior to performing three 20-m sprints at 4, 8 and 12 min. In week 4, the same sessions as in week 3 were repeated, so there was a total of four experimental sessions performed over a two-week period. All participants were instructed not to perform any exercise at least 24 h prior to all sessions. Participants were also instructed to abstain from caffeine and alcohol consumption for at least 12 h before all sessions. These precautions were taken to minimize the effect on the outcome measures. An experienced exercise scientist supervised all sessions.

Each experimental session commenced with a 10-min standardized warm-up which incorporated light jogging, dynamic stretching, tuck jumps and sprinting. Dynamic stretching mainly focused on the lower body muscles utilized in sprinting and included standing forward leg swings, walking lunges, and internal and external hip rotation. Following the stretches three 20-m sprints (two minutes rest between bouts) were performed progressing to maximal effort on the last sprint.

For the KBS condition, participants performed 2 sets of 5 KBS using a modified apparatus of the kettlebell, the Kettle Clamp (Kettle Clamp, Ohio, USA). Briefly, dumbbells are attached to the kettle clamp which allows for a greater weight range to be used; otherwise, limited by standard kettlebell weights. Participants completed each set of KBS within one minute, with three minutes of passive rest incorporated between sets; therefore, the total duration of the KBS exercise was five minutes. After completion of the KBS, participants rested in a standing position for four minutes prior to performing the first 20-m sprint. The 4-min rest period was selected due to previous literature describing moderately trained individuals requiring 3–7 min of recovery to offset the level of fatigue after a heavy resistance warm-up, in order to produce a potentiating response [16,36]. Subsequent sprints were performed at 8- and 12-min post KBS, again with participants resting in a standing position between bouts. For the CON condition, after participants completed their warm-up, they rested in a standing position for nine minutes prior to the first sprint. The rest period prior to the first 20-m sprint for the CON condition matched the time between completion of the warm-up and the first sprint during the KBS condition (i.e., KBS duration of 5 min plus 4 min passive rest). Participants then performed sprints at 13- and 17-min post warm-up. Therefore, sprints were performed 4, 8 and 12 min post CON condition. The best sprint performances to 5-m, 10-m and 20-m from the KBS and CON sessions were used for data analysis.

All 20 participants completed the preliminary sessions and the week 3 experimental sessions. However, there were technical issues in week 3 that led to data not being recorded for one of the three sprints for four participants. Therefore, the week 4 data for these sprints and not the best of two performances were used for these participants. Also, in week 4, one male participant was unable to complete both the KBS and CON sessions and one female participant could not complete the KBS session (both due to unrelated musculoskeletal injuries). For these two participants the week 3 results were used and not the best of the two experimental weeks. The test-retest reliability between sessions was therefore assessed on sample size n ≥ 16. 

### 2.4. Statistical Analysis

All data were inspected visually and statistically for normality using the Kolmogorov–Smirnov test and were found to be normally distributed. Data are presented as mean ± standard deviation. Descriptive statistics of males and females including body composition (lean body and fat mass) and muscle performance (leg extensor strength and power, KBS optimal load) were reported to describe fitness-related characteristics of the participants in this study. Test-retest reliability (i.e., sessions in week 3 versus week 4) was calculated using intraclass correlation coefficients (ICC) and assessed using the scale < 0.5 poor, 0.5–0.75 moderate, >0.75–0.9 good, and >0.90 excellent reliability. Effects of KBS on sprint time were analyzed using a two factor repeated measures ANOVA, one factor being time (with three levels: 4, 8, and 12 min) and the other being condition with two levels corresponding to the two conditions (KBS and CON). For significant ANOVA results, a post hoc Tukey’s test was used to determine which group means differed. Effect Sizes (ES) between the two conditions were calculated using Morris and Deshon’s equation 8 [37]. The magnitude of the ES were assessed using the following criteria; trivial: ≤ 0.2, small: 0.21–0.59, moderate: 0.6–1.19, large: 1.2–1.99, very large: 2.0–3.9, and extremely large: ≥ 4.0 [38]. Percent (%) changes for each condition were calculated using the following formula: % change = ((CON value minus KBS value) divided by CON value) multiplied by 100.

Partial correlation analyses (adjusting for sex) were performed to examine relationships between change in sprint performance (difference between KBS and CON) and muscle performance characteristics (relative load used for KBS, relative isometric strength, and isokinetic power at 90 °/s and 250 °/s). Strength of correlations were qualitatively assessed using the following criteria: trivial (r < 0.1), small (r > 0.1 to 0.3), moderate (r > 0.3 to 0.5), strong (r > 0.5 to 0.7), very strong (r > 0.7 to 0.9), nearly perfect (r > 0.9), and perfect (r = 1.0) [38]. All analyses were performed using SPSS Statistics Version 24 for Windows, with significance level set at p < 0.05. 

## 3. Results

### 3.1. Effect of Kettlebell Swings versus Control on Sprint Time

Sprint time at 5-m, 10-m and 20-m following the three respective recovery times (4, 8, and 12 min) for the two conditions (KBS versus CON) is displayed in Table 2. There were no condition effects, time effects or condition by time interactions for sprint times at 5-m and 10-m. At 5-m, small ES in favor of CON were found at all recovery times (ES = −0.10 to −0.19), whereas at 10-m trivial ES between conditions were found at all recovery times (ES = −0.02 to −0.08). A significant time effect was found for sprint time at 20-m (*p* = 0.042), with the post hoc analysis revealing a faster sprint time at 12 min compared to 4 min (*p* = 0.022). There was no condition effect or condition by time interaction found for sprint time at 20-m. Trivial ES between conditions at 20-m were found at all recovery times (ES = −0.02 to −0.04). Intra Class Correlations between the two experimental weeks showed that reliability was good to excellent (ICC = 0.77–0.97).

### 3.2. Effect of Muscle Performance Characteristics on PAP

Table 3 displays the relationship between change in sprint time (i.e., CON minus KBS sprint time) following conditions and muscle performance characteristics. There were moderate negative correlations found for change in sprint time and KBS load (absolute and relative) at 4 min recovery (r = −0.31 to −0.42). This finding suggested that participants with a greater KBS load (absolute and relative) had a faster sprint time at 4 min following the CON compared to KBS condition. At 8- and 12-min recovery time there were small to moderate positive correlations found for change in sprint time and KBS load (absolute and relative) (*r* = 0.21 to 0.37). This result suggests faster sprints at 8 and 12 min following KBS for participants with greater KBS load (absolute and relative). Trivial to small correlations between change in sprint time and all other muscle performance measures (i.e., isometric strength, isokinetic power at 90 °/s and 250 °/s) were found at 5-m, 10-m and 20-m for all recovery times.

## 4. Discussion

This study investigated whether including kettlebell swings (KBS) in a warm-up enhances 20-m sprint performance. To our knowledge, this is the first study to incorporate KBS as a preconditioning exercise prior to sprint running. The findings from the present study showed no potentiating effect of KBS on sprint time. Initial sprint time over 20-m was found to improve following further attempts regardless of whether a KBS were included in the warm-up or not. The small to moderate positive correlations found between change in sprint time and KBS load suggests that people with the ability to lift heavier loads for the KBS may require at least 8 min recovery, if any potentiation effect is to result. Furthermore, including KBS in a warm-up may be detrimental to sprint time for people stronger with this exercise if performed 4 min or less before a sprint. Therefore, while it appears that including KBS in a warm-up does not influence subsequent sprint performance, it may be dependent on individual strength capabilities with the KBS. 

Previous research investigating the acute effects of high resistance exercises on sprint performance have used barbell squats [6,9,10,25], with one study incorporating power cleans [7]. However, the dynamics of the KBS are thought to be more movement specific to sprint running than barbell squats, where there is greater horizontal propulsive forces and higher activation of the muscles (hamstrings and gluteals) incorporated in sprint running. In addition, a study by Manocchia et al. [39] found KBS to transfer effectively to Olympic style lifts as well as other explosive movements. This was due to the KBS performed rapidly, which led to increased maximal power output and increased rate of force development. For the present study, it seemed plausible that due to KBS being performed rapidly, an increased activation of the hamstrings and gluteal muscles, producing high forces and velocities (similar to barbell squats and power cleans) would acutely enhance sprint performance. Results from the current study showed that the implementation of KBS as a preconditioning exercise did not improve sprint time at any distance and recovery time. This finding may be attributed to the study design that was implemented (i.e., exercise prescription) and characteristics of the participants in the present study (e.g., individual strength, training experience, type 2 muscle fiber distribution) [13,16,17,18].

A lack of improvement in acute exercise performance following a preconditioning exercise is thought to be linked to the coexistence of potentiation and fatigue within the contracting muscle [5,40]. The majority of research on PAP have reported improvements in sprint performance following recovery times ranging from 4–12 min [7,9,10], with two studies reporting the greatest potentiating effect to occur between 7–10 min [8,25]. Based on this evidence, rest intervals of 4, 8 and 12 min seemed to be adequate to enable the potentiating effect from the KBS to dissipate at a slower rate than the effect of fatigue [40]. Furthermore, 4–5 min is required between explosive performance to replenish immediate fuel stores [12,40]. However, similar to the findings of the present study, no reductions in sprint time were found at 4, 8 and 12 min following a 3RM barbell squat in a group of sub elite rugby players. Therefore, it is possible that the strength levels of participants in the previous and present study may have contributed to the absence of a potentiating stimulus [16,17,23]. In the present study, the participants were recreationally trained individuals that were participating in various sports. In contrast, the studies that have found a potentiating effect on sprint performance have been in professional rugby players, and track and field athletes, who are known to possess greater relative strength compared to sub-elite sportspeople [8,23,24]. Another plausible explanation is that other exercises performed during the warm-up (e.g., jumps/sprints) provided similar enhancement as the KBS and potentially masked the KBS effect.

Previous research has also found that athletes involved in explosive sports tend to exhibit a greater potentiating effect than recreationally trained individuals [13,16,17]. This is highlighted in one study [13] where athletes increased their countermovement jump height after a 5RM barbell squat, as compared to recreationally trained individuals whom experienced a decline in countermovement jump height. It could also be hypothesized that individuals with greater power output capabilities may have greater PAP potential. This may be supported by the present study findings of small to moderate correlations found for KBS (absolute and relative loads) and sprint performances at 8 and 12 min. Whereas, strength measured isometrically and at different velocities from the knee extensors does not appear to relate to the ability to potentiate sprint performance from KBS based on the present study findings of trivial to small correlations. Therefore, it appears that individuals who were stronger with the KBS experienced a greater improvement in sprint performance following the KBS condition (i.e., greater PAP effect). 

Multiple sets (≥ 2) at a moderate intensity have been shown to be more optimal for potentiation than a single set protocol [17], although a multiple set protocol should also theoretically produce more fatigue than a single set [16]. Also, an individual’s strength level has been shown to effect PAP from single and multiple set protocols [17], with stronger individuals benefiting from multiple set protocols while weaker individuals benefited from single sets. Since the participants in the present study were recreationally trained, it is possible that the prescription of multiple sets of KBS as a preconditioning stimulus for the 20-m sprint was not suitable. However, since sprint performance measures were quite similar at all recovery times for both the KBS and CON conditions, it seems unlikely that fatigue as a result of the KBS condition (i.e., 2 sets of 5 repetitions) negatively influenced the results.

A strength of the current study was the good to excellent reliability between the two experiment weeks for the KBS and CON conditions. As such, there is increased confidence in the results showing that KBS do not acutely enhance 20-m sprint performance in recreationally trained adults. It is possible that the KBS may potentiate other activities such as the vertical jump, which is another activity commonly investigated in PAP research [5,15] and should be investigated in future research.

There are several limitations that need to be acknowledged when interpreting the findings of the present study. The technique used for the KBS needed to be slightly modified when a heavier load was used. Rubber hexagonal dumbbells were attached to the kettle clamp and as the load increased so did the width of the dumbbell, which resulted in stronger participants adopting a wider stance to remain balanced during the KBS. This wider stance used by the stronger participants may have reduced the ability of the hips and hamstrings to develop maximum power throughout the movement, with greater focus on maintaining a balanced position. Additionally, there is no function on the kettle clamp to adjust the orientation of the dumbbell where the rubber hexes are facing anteriorly and posteriorly. This would have allowed participants with higher optimal loads to perform the swings without modification of their technique. Finally, the inclusion of both males and females as participants may have contributed to the mixed findings due to sex differences in relative strength. However, we did consider sex differences when analysing relationships with muscle performance characteristics through running partial correlation analyses that were adjusted for sex. 

## 5. Conclusions

The findings from the present study indicate that the KBS does not potentiate sprint performance. Therefore, it is not recommended that athletes perform KBS as a preconditioning exercise for the potentiation of the 20-m sprint. It is possible that these results were influenced by individual strength capabilities. Since this is the first known study to investigate the effect of KBS as a preconditioning exercise on sprint performance, further research is required to substantiate the current study’s findings.

## Figures and Tables

**Figure 1 sports-07-00036-f001:**
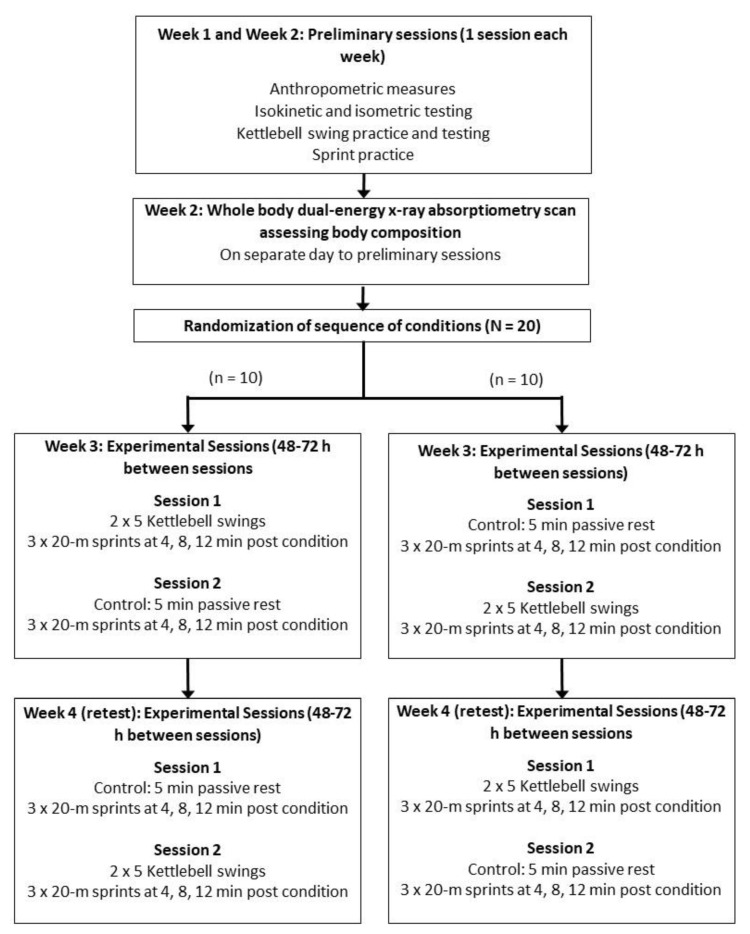
Diagrammatic representation of the study design.

**Table 1 sports-07-00036-t001:** Anthropometric and Muscle Performance Characteristics of Participants.

Variable	Males (*n* = 12)	Females (*n* = 8)
Age (y)	21.7 ± 2.3	21.9 ± 2.0
Height (cm)	178.3 ± 7.0	159.2 ± 4.4
Body Mass (kg)	74.5 ± 8.3	54.1 ± 5.2
Fat Mass (kg)	9.2 ± 6.3	12.2 ± 3.0
Body Fat (%)	12.4 ± 7.1	23.4 ± 4.0
Lean Body Mass (kg)	62.7 ± 6.0	39.5 ± 3.2
Optimal Load for KBS (kg)	42.5 ± 14.0	27.3 ± 3.8
Isometric LE Strength (Nm)	308.6 ± 66.6	209.6 ± 50.4
Isokinetic LE Power at 90 °/s (W)	431.5 ± 84.7	268.8 ± 74.1
Isokinetic LE Power at 250 °/s (W)	1030.1 ± 236.1	685. 6 ± 170.3

KBS = kettlebell swing; Optimal Load for KBS = load that achieves the highest mean power for the KBS; LE = leg extension.

**Table 2 sports-07-00036-t002:** Sprint time at 4-, 8- and 12-min following kettlebell swing and control conditions.

Distance (m)	Condition	4 min	% Change (ES)	8 min	% Change (ES)	12 min	% Change (ES)	ANOVA (*p*)		
								C	T	C × T
5	KBS	1.12 ± 0.10	2.2 (−0.19)	1.11 ± 0.12	1.4 [−0.10]	1.11 ± 0.09	1.3 (−0.14)			
	CON	1.10 ± 0.12		1.10 ± 0.12		1.09 ± 0.09		0.204	0.924	0.780
10	KBS	1.90 ± 0.17	0.8 (−0.08)	1.90 ± 0.18	0.3 (−0.02)	1.89 ± 0.15	0.4 (−0.05)			
	CON	1.89 ± 0.17		1.90 ± 0.18		1.88 ± 0.15		0.528	0.217	0.860
20	KBS	3.31 ± 0.31	0.4 (−0.04)	3.30 ± 0.33	0.2 (−0.02)	3.29 ± 0.32	0.3 (−0.04)			
	CON	3.30 ± 0.31		3.29 ± 0.32		3.28 ± 0.30		0.545	0.042 *	0.921

* Significant at *p* < 0.05.; C = condition effect; C × T = condition x time interaction; CON = control; ES = effect size; KBS = kettlebell swing; m = metres; min = minutes; T = time effect. A positive % change indicates a slower time for KBS compared to CON.

**Table 3 sports-07-00036-t003:** Correlation matrix between change in sprint time (control minus kettlebell swing conditions) and muscle performance characteristics.

Variable	KBS Load	Relative KBS Load	Isometric LE Strength	Relative Isometric LE Strength	Isokinetic LE Power at 90 °/s	Relative Isokinetic LE Power at 90 °/s	Isokinetic LE Power at 250 °/s	Relative Isokinetic LE Power at 250 °/s
Sprint (4 min post)								
5-m	−0.42	−0.39	−0.11	−0.05	0.09	0.12	0.05	0.07
10-m	−0.41	−0.38	−0.08	−0.01	−0.01	0.04	0	0.03
20-m	−0.36	−0.31	−0.08	0	0.05	0.13	0.05	0.11
Sprint (8 min post)								
5-m	0.37	0.34	0.04	−0.04	0.14	0.06	0.10	−0.01
10-m	0.27	0.25	−0.03	−0.10	0.11	0.05	0.08	0
20-m	0.21	0.25	−0.18	−0.21	0.06	0.08	0.10	0.08
Sprint (12 min post)								
5-m	0.22	0.35	−0.26	−0.19	0.01	0.11	0.17	0.23
10-m	0.23	0.29	−0.04	−0.02	0.13	0.14	0.19	0.16
20-m	0.23	0.29	−0.09	−0.07	0.14	0.18	0.22	0.21

* Correlation is significant at *p* < 0.05. Relative = Expressed relative to body mass. KBS = kettlebell swing; LE = leg extension; min = minutes. Positive correlation = faster sprint time from KBS correlated with muscle performance measure.

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
