# Peer review of "Acute Effect of Kettlebell Swings on Sprint Performance"

_sports, 2019, doi:10.3390/sports7020036_

Round 1
Reviewer 1 Report
General Comments:
The authors have provided a novel manuscript examining the acute effects of kettlebell swings on sprint performance. The manuscript is generally well-written; however, some grammar improvements can be made. In addition, the authors should re-consider the use of the term 'ballistic' to describe kettlebell swings. I would agree that they are performed rapidly, but they are not truly ballistic. If they were, the participant would leave the ground or throw the kettlebell. More specific comments are below.
Specific Comments:
Lines 27-29: The term 'studies' often refers to many studies. The authors have cited two here which is a good start; however, additional literature is needed. Consider using some of the references cited later in the manuscript to supplement the current ones.
Lines 98-100: What knee angle was used for the assessment of isometric peak torque and what was the rationale for using that angle?
Line 103: Add a comma after 'Prior to testing...'
Lines 114-115: A figure showing the sequence of the exercise would be helpful to the reader.
Line 116: While I certainly agree that there are horizontal forces being produced during short sprints, the authors should keep in mind that the longer the sprint gets the more dominant vertical forces become. In fact, some research suggests that it is about 50/50 vertical and horizontal coming out of the blocks. That being said, I would caution the authors in believing that primarily horizontal forces are THE forces individuals emphasize during short sprints and kettlebell swings.
Line 120: As mentioned above, the authors should reconsider the use of the term ballistic here. Did the subjects leave the ground during the KBS? Ballistic essentially refers to accelerating throughout the entire motion to the point where the limbs lose contact with the ground or the object. That being said, KBS are typically classified as semi-ballistic movements primarily because the individual does not jump off of the ground nor do they create a projectile by throwing the kettlebell. Are they performed with maximal intent? Yes; however, not to the point where they become truly ballistic.
Line 129: Add a comma after '...output detected..'
Line 136: Change 'offset' to 'trigger'.
Line 143: I do not recall the reliability results of the current study being presented. The authors must include these data for the current study.
Lines 166-172: To be clear, some of their data is currently included, but some of it is not? It would not be appropriate to only include some of their data if other data is missing within the comparisons. This alters the n used for each comparison. Further clarification is needed here as this may alter the overall results of the study.
Table 2: It is unclear what numbers are actually in the brackets. Please clarify.
Line 242: Going along with this, what was the rationale of assessing knee extensor strength rather than knee flexion strength?
Lines 244 and 246: See comment above regarding the use of the term 'ballistic'.
Line 261: Change 'was' to 'were'.
Line 263: Change 'this' to 'the'.
Line 264: Add a comma after 'In the present study'
Line 278: Consider modifying this sentence to read '...relate to the ability to...' since you assessed the relationships in this study.
Lines 291-305: Another potential limitation that was not addressed was the use of both men and women as participants. It is possible that the relative strength of the sexes may have been vastly different ultimately contributing to mixed findings. The authors should add some content within their limitations about this.
Lines 300-305: The authors can certainly include these sentences in the discussion; however, the current placement within the limitations is confusing.
Line 307: Maintain the use of your KBS acronym here.
Author Response
We thank the reviewer 1 for their constructive comments which have enabled us to improve the manuscript. Please find below a point-by-point response to all the comments raised.
REVIEWER 1
Comment 1: The authors should re-consider the use of the term 'ballistic' to describe kettlebell swings. I would agree that they are performed rapidly, but they are not truly ballistic. If they were, the participant would leave the ground or throw the kettlebell. More specific comments are below.
Response 1:
All instances where kettlebell swings were referred to as ‘ballistic’ have now been changed so that it is emphasised they are ‘performed rapidly’.
Comment 2:
Lines 27-29: The term 'studies' often refers to many studies. The authors have cited two here which is a good start; however, additional literature is needed. Consider using some of the references cited later in the manuscript to supplement the current ones.
Response 2:
Extra references have been added.
Comment 3:
Lines 98-100: What knee angle was used for the assessment of isometric peak torque and what was the rationale for using that angle?
Response 3:
The knee angle was 90 degrees flexion and this information has been added to the methodology. This is the standard knee angle used in our laboratory for assessing MVIC of the knee extensors. We do not feel that any rationale for the knee angle used to assess MVIC of the extensors is required in the manuscript.
Comment 4:
Line 103: Add a comma after 'Prior to testing...'
Response 4:
This has now been corrected.
Comment 5:
Lines 114-115: A figure showing the sequence of the exercise would be helpful to the reader.
Response 5:
A figure detailing the study design has been added.
Comment 6:
Line 116: While I certainly agree that there are horizontal forces being produced during short sprints, the authors should keep in mind that the longer the sprint gets the more dominant vertical forces become. In fact, some research suggests that it is about 50/50 vertical and horizontal coming out of the blocks. That being said, I would caution the authors in believing that primarily horizontal forces are THE forces individuals emphasize during short sprints and kettlebell swings.
Response 6:
Yes we totally agree, however as you have rightly acknowledged horizontal forces influence short sprint performance and we have selected a version of the kettlebell swing that allows for better generation of these forces. We have not suggested that horizontal forces play the major role in short sprint performance.
Comment 7:
Line 120: As mentioned above, the authors should reconsider the use of the term ballistic here. Did the subjects leave the ground during the KBS? Ballistic essentially refers to accelerating throughout the entire motion to the point where the limbs lose contact with the ground or the object. That being said, KBS are typically classified as semi-ballistic movements primarily because the individual does not jump off of the ground nor do they create a projectile by throwing the kettlebell. Are they performed with maximal intent? Yes; however, not to the point where they become truly ballistic.
Response 7:
All instances where kettlebell swings were referred to as ‘ballistic’ have now been changed so that it is emphasised they are ‘performed rapidly’.
Comment 8:
Line 129: Add a comma after '...output detected..'
Response 8:
This has been corrected.
Comment 9:
Line 136: Change 'offset' to 'trigger'.
Response 9:
This has been changed.
Comment 10:
Line 143: I do not recall the reliability results of the current study being presented. The authors must include these data for the current study.
Response 10:
The reliability has been reported in lines 210-211 and is also shown below.
“Intra Class Correlations between the two experimental weeks showed that reliability was good to excellent (ICC = 0.77-0.97).”
Comment 11:
Lines 166-172: To be clear, some of their data is currently included, but some of it is not? It would not be appropriate to only include some of their data if other data is missing within the comparisons. This alters the n used for each comparison. Further clarification is needed here as this may alter the overall results of the study.
Response 11:
Your interpretation is incorrect because all subjects had data for all conditions and sprints and this was used during the statistical analysis to determine the effect of KBS on sprint performance. However, not all subjects had TWO SETS of data for all conditions and sprints, therefore this affected the number of subjects used to assess the reliability of measures. This information has been included in the Methods (shown below).
“The test-retest reliability between sessions was therefore assessed on sample size n = ≥ 16.”
Comment 12:
Table 2: It is unclear what numbers are actually in the brackets. Please clarify.
Response 12:
‘ES’ is already stated within the brackets and the legend defines this as ‘Effect Size’.
Comment 13:
Line 242: Going along with this, what was the rationale of assessing knee extensor strength rather than knee flexion strength?
Response 13:
We wanted to assess both knee extensor and flexion strength (and had ethics approval for both) however due to time restraints we could only assess one of these and decided to go with the former.
Comment 14:
Lines 244 and 246: See comment above regarding the use of the term 'ballistic'.
Response 14:
This has been amended throughout the manuscript so that KBS is now referred to as “being performed rapidly” rather than being described as a “ballistic” exercise.
Comment 15:
Line 261: Change 'was' to 'were'.
Response 15:
This has been changed.
Comment 16:
Line 263: Change 'this' to 'the'.
Response 16:
This has been amended.
Comment 17:
Line 264: Add a comma after 'In the present study'
Response 17:
This has been added.
Comment 18:
Line 278: Consider modifying this sentence to read '...relate to the ability to...' since you assessed the relationships in this study.
Response 18:
This has been amended.
Comment 19:
Lines 291-305: Another potential limitation that was not addressed was the use of both men and women as participants. It is possible that the relative strength of the sexes may have been vastly different ultimately contributing to mixed findings. The authors should add some content within their limitations about this.
Response 19:
The following has been added to the limitations.
“Finally, the inclusion of both males and females as participants may have contributed to the mixed findings due to sex differences in relative strength. However, we did consider sex differences when analysing relationships through running partial correlation analyses that were adjusted for sex.”
Comment 20:
Lines 300-305: The authors can certainly include these sentences in the discussion; however, the current placement within the limitations is confusing.
Response 20:
We agree and have moved these sentences above the limitations section.
Comment 21:
Line 307: Maintain the use of your KBS acronym here.
Response 21:
This has been corrected.
We trust that the issues above have been addressed and clarified sufficiently and we look forward to hearing from you in the near future.
Sincerely,
Dr Daniel Hackett
Reviewer 2 Report
Please find comments in the attached file

Author Response
We thank the reviewer 2 for their constructive comments which have enabled us to improve the manuscript. Please find below a point-by-point response to all the comments raised.
REVIEWER 2
Comment 1:
More profound overview of the previous studies one effects of different PAP protocols is needed. Authors only sporadically noted that PAP protocols were found as being effective, but more details about studies where investigators observed similar participants and capacities is needed. Otherwise, it seems that you arbitrary decided to study the problem you studied.
Response 1:
We agree and have expanded the information within the Introduction so that the reader can understand the rationale for this study (shown below).
“There have been two main theories proposed to explain an enhancement in athletic pursuits from PAP. These include 1) increased stimulation of the central nervous system, after preconditioning exercise, prior to an explosive activity [11, 12] and 2) increased phosphorylation of myosin regulatory light chains within muscle fibres [12]. Both these theories explain an increase in contraction velocity of the potentiating muscle, thus leading to an increase in its power output during an explosive activity such as jumping, throwing, and sprinting.”
“Contrary to the findings of PAP effects on subsequent exercise performance, some studies have reported that acute exercise performance did not change [13,14] or was impaired [15] following a preconditioning exercise. An explanation for the conflicting results is likely due to the coexistence of potentiation and fatigue within the contracting muscle.”
“This suggests that for stronger individuals an increase in PAP from single to multiple sets outweighs the increase in fatigue. Therefore, individual characteristics (e.g. muscular strength and training experience) and various aspects of the exercise protocol used such as rest period, intensity and volume may influence the PAP effect from preconditioning exercises [12].”
”Previous studies that have investigated the effects of preconditioning exercises on sprint performance have used ballistic exercises such as depth jumps and alternate leg bounding [20,21] or resistance exercises such as barbell squats and power cleans [6-10,22-25 ]. The majority of these studies have shown improved sprint performance following the ballistic exercises [20,21] and resistance exercises [6-10,25]. However, it remains uncertain which type of exercise is more effective for maximising the potentiating effect on sprinting [2,16]. Whereas the ballistic exercises require minimal equipment (e.g. weight vest) or no equipment, the performing of resistance exercises requires specialised equipment (e.g. barbells, weight plates, squat rack), which is generally inaccessible immediately prior to competitions [2,23].”
Comment 2:
Methods Experimental design - This is a relatively complex study and experimental design should be presented more concisely. Actually, I had to read this paragraph several times in order to understand it properly (and still not sure that I understood it in details). I strongly suggest you to divide "familiarization sessions" from "experimental sessions" (even use some subheadings when explaining)
Response 2:
We agree and have divided “preliminary sessions: from experimental sessions.
Comment 3:
Variables - Tests should be presented as separate paragraph, and then explain each specific test without numbering it as subheading.
Response 3:
We have made these amendments are suggested.
Comment 4:
Statistics – Start with descriptive statistics (what did you calculate and why? Any testing of normality?). The fact that you used best performances for sprint-time should be noted in the Methods section and not here. Then move to reliability (it should be done before other analyses), then to ANOVA effects and ESs and % changes, and correlations at the end.
Response 4:
We have made these suggested changes.
Comment 5:
Results
This is the weakest point of your paper. First, you should present ANOVA results, while I can’t see that you mentioned it. Am I missing something? Table 2 is not well prepared (see bold numbers and underlining?) You have measured body composition, and did not use it in analyses. OK, you have two genders, but then again – why it was important to obtain data on body composition?
Response 5:
The ANOVA results have now been included in Table 2.
The bold numbers and underlining has been corrected.
Body composition was assessed to describe the characteristics (which is fitness-related) of the participants in the study.
We trust that the issues above have been addressed and clarified sufficiently and we look forward to hearing from you in the near future.
Sincerely,
Dr Daniel Hackett
Round 2
Reviewer 1 Report
The authors have adequately addressed all of my concerns.
Reviewer 2 Report
Thank you for following my comments and suggestions